# Functional diversity of excitatory commissural interneurons in adult zebrafish

**E Rebecka Björnfors, Abdeljabbar El Manira\***

Department of Neuroscience, Karolinska Institute, Stockholm, Sweden

**Abstract** Flexibility in the bilateral coordination of muscle contraction underpins variable locomotor movements or gaits. While the locomotor rhythm is generated by ipsilateral excitatory interneurons, less is known about the commissural excitatory interneurons. Here we examined how the activity of the V0v interneurons – an important commissural neuronal class – varies with the locomotor speed in adult zebrafish. Although V0v interneurons are molecularly homogenous, their activity pattern during locomotion is not uniform. They consist of two distinct types dependent on whether they display rhythmicity or not during locomotion. The rhythmic V0v interneurons were further subdivided into three sub-classes engaged sequentially, first at slow then intermediate and finally fast locomotor speeds. Their order of recruitment is defined by scaling their synaptic current with their input resistance. Thus we uncover, in an adult vertebrate, a novel organizational principle for a key class of commissural interneurons and their recruitment pattern as a function of locomotor speed.

## Introduction

The precise coordination of movements on the two sides of the body is an essential feature of locomotion in animals with bilateral symmetry (*Grillner and Jessell, 2009*; *Arber, 2012*; *Moult et al., 2013*; *Kiehn, 2016*). The left-right coordination during locomotion can vary in a context-dependent manner to produce alternating, e.g. walking in limbed animals or swimming in fish, or synchronous, e.g. hopping movements in limbed animals. This is largely achieved by regulating the activity of populations of commissural interneurons connecting local interacting circuits on the two sides of the spinal cord (*Soffe et al., 1984*; *Dale, 1985*; *Buchanan, 1999a*, *1999b*; *Butt and Kiehn, 2003*). The V0 interneurons, which originate from the p0 progenitor domain, represent a major commissural neuronal population controlling the left-right alternation of locomotor movements. The molecular mechanisms of differentiation of the V0 interneurons display striking similarities between zebrafish and mice. This interneuron population can be subdivided into two broad classes; one is excitatory and located ventrally early during development (V0v), and the other is inhibitory and occupies a dorsal position (V0d) (*Moran-Rivard et al., 2001*; *Pierani et al., 2001*; *Lanuza et al., 2004*). In addition, a detailed birth-date analysis in zebrafish has revealed three different subclasses of excitatory V0v interneurons arising in an order that correlates with their morphology and axonal projections, arguing for a further subdivision of this class of interneurons (*Satou et al., 2012*).

The current view of commissural V0 function in mediating the left-right alternation of locomotor movements derives from studies in immature motor systems (newborn mice and zebrafish larvae) and without detailed information about the activity at the single neuron level during locomotion. In newborn mice, a pattern of recruitment of the V0 interneurons has been proposed based on results from genetic ablations of the inhibitory (V0d) or the excitatory (V0v) interneurons. The two classes of V0 interneurons are thought to be differentially recruited to mediate the left-right alternation at

**\*For correspondence:** Abdel.
Elmanira@ki.se

**Competing interests:** The authors declare that no competing interests exist.

**eLife digest** During movements such as swimming and walking, the left and right sides of the body are kept coordinated by specific neurons in the spinal cord. Some of these neurons – called V0 neurons – can either excite or inhibit neurons on the opposite side of the spinal cord. In mice, the inhibitory V0 neurons are responsible for left-right coordination when the mice are moving slowly, while the excitatory neurons operate when the animals are moving more quickly. However, in zebrafish larvae a group of excitatory V0 neurons are only active when the larvae are swimming slowly.

Björnfors and El Manira investigated whether excitatory V0 neurons in adult zebrafish behave like those in the larvae, or whether they act more like those in mice. The experiments show that the excitatory V0 neurons in adult zebrafish can be separated into three groups that are activated either at slow, intermediate or fast speeds of movement. The activation of the excitatory V0 neurons depends on the properties of the neurons themselves in combination with signals they receive from other neurons in the spinal cord.

Although the excitatory V0 neurons could be active across all speeds, Björnfors and El Manira found that more neurons were active at faster speeds. This suggests that, in the adult zebrafish, there are both similarities and differences in how the V0 neurons are organised compared to larval zebrafish and mice. The next step following on from this work would be to find out the specific roles of excitatory and inhibitory V0 neurons during movement.

different locomotor speeds. The inhibitory V0d interneurons are recruited first, during slow locomotion, and mediate direct inhibition. The excitatory V0v interneurons are recruited only at higher frequencies and mediate alternation by activating contralateral inhibitory interneurons (*Talpalar et al., 2013*; *Bellardita and Kiehn, 2015*). By contrast, in larval zebrafish, a morphologically defined subclass of V0v interneurons, the multipolar commissural descending (MCoD), has been shown to be recruited only at slow swimming speeds and become inhibited at faster swimming frequencies (*McLean et al., 2007*; *McLean and Fetcho, 2009*). The apparent discrepancy between the proposed roles of V0v interneurons in zebrafish versus in mice could be the result of a functional switch during evolution that could account for the difference in the speed-dependent recruitment of the V0v population in these two species. Alternatively, the V0v interneurons could represent a functionally heterogeneous population with separate subclasses being engaged with increased locomotor speed. While there is mounting evidence for such a heterogeneity within the spinal interneuron classes (e.g. V2a and V1) (*Kimura et al., 2008*; *McLean and Fetcho, 2009*; *Ausborn et al., 2012*; *Satou et al., 2012*; *Griener et al., 2015*; *Bikoff et al., 2016*), a direct characterization of V0v interneuron diversity has remained elusive.

Here, we have examined the functional heterogeneity of V0v interneurons in the adult zebrafish spinal cord, a feat so far intractable in the adult mouse. We sought to determine if the V0v interneurons form one class engaged only within a specific swimming frequency range or if they are subdivided into functional sub-classes differentially recruited at specific speeds. The results show that this is a functionally heterogeneous sub-class of interneurons composed of three sub-types recruited and active at specific speeds during the swimming motor program, and one additional sub-type that does not display any locomotor-related activity. Furthermore, we show that the V0v interneurons which are active during locomotion are engaged at slow, intermediate or fast swimming frequencies and that their recruitment thresholds are determined by the scaling of their synaptic drive and intrinsic properties. Our work thus reveals that the V0v interneurons are not homogenously engaged at a defined locomotor speed, but conform to a modular organization that allows them to participate in the elaboration of the locomotor pattern at all speeds.

## Results

### Distribution of V0v interneurons in the spinal cord

The distribution of the V0v interneurons in the spinal cord of adult zebrafish was examined by crossing the Tg[*vglut2*:loxP-DsRed-loxP-GFP] with the Tg[*dbx1b*:Cre] line which results in GFP expression in the excitatory V0v interneurons (*Satou et al., 2012*, *2013*). These interneurons have axons that initially project ventrally from the somata before they cross the midline (*Figure 1A,B*). The axonal projections of the V0v interneurons were carefully examined in preparations mounted with the dorsal side up (*Figure 1C*). These interneurons first project their axons laterally before turning to cross the midline (*Figure 1C*). The V0v interneurons were evenly distributed along the rostro-caudal axis in the spinal cord (*Figure 1D*). These interneurons were also distributed along the dorso-ventral axis with a large proportion occupying the middle of the spinal cord (*Figure 1D*). There were 21 ± 6 V0v interneurons per hemi-segment (n = 6 preparations). The distribution of V0v interneurons in the spinal cord as well as their number per hemi-segment are comparable to those reported for the excitatory V2a interneurons (20 per hemi-segment), while motor neurons are more numerous (40 per hemi-segment) (*Ampatzis et al., 2013*).

### Activity pattern of glutamatergic V0 interneurons during locomotion

The available data from larval zebrafish and newborn mouse suggest that the V0v interneurons are not active at all speeds of locomotion but are only engaged during slow or fast speeds in zebrafish and mice, respectively. However, in larval zebrafish, recordings were only limited to a single morphologically defined sub-class of V0v interneurons (MCoDs), while in mice the V0v interneuron activity has thus far not been possible to evaluate during locomotion. Therefore, we sought to determine the pattern of activity of the V0v interneurons during the swimming motor pattern in adult zebrafish. To this end, we used the ex-vivo brainstem-spinal cord preparation in which fictive swimming activity is induced by stimulation of descending inputs and monitored by recording from a motor nerve

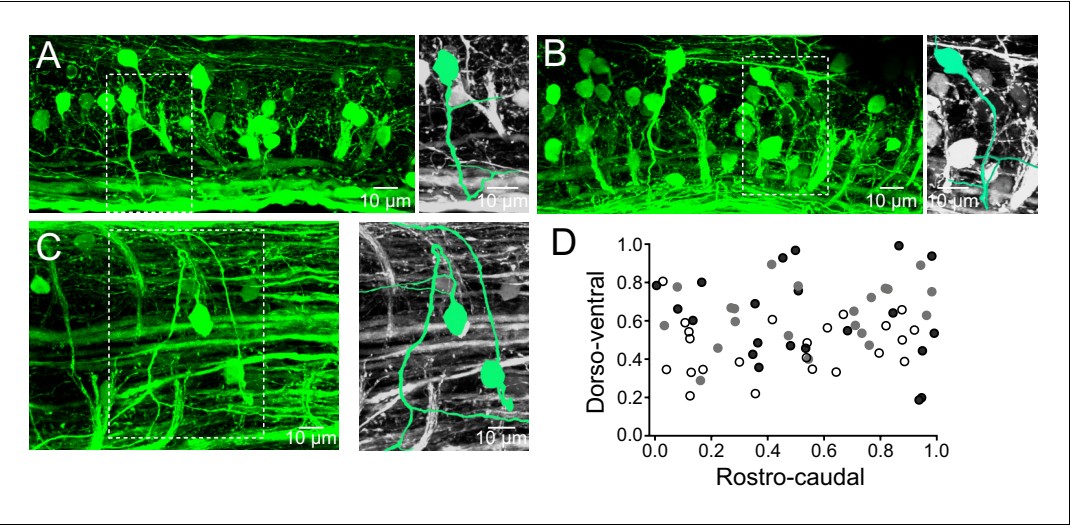

**Figure 1.** Distribution of V0v interneurons in the spinal cord. (A,B) Lateral views of a spinal segment from two different preparations showing the distribution of GFP-expressing V0v interneurons. Images are the maximum intensity projections from 164 and 193 optical sections respectively. Most V0v interneurons project ventrally before crossing the midline as indicated in expansions (right panels). (C) Dorsal view of the spinal cord showing two V0v interneurons and their crossing axonal projections (neurons are colored and expanded in the right panels for clarification). The image is the maximum intensity projections from 90 optical sections (D) Graph showing the dorso-ventral and rostro-caudal distribution of V0v interneurons within one hemisegment. Data are from three different preparations (black, grey and open circles). The ventral edge is defined as 0 and the dorsal edge as 1. V0v interneurons are evenly distributed along the rostro-caudal axis, with an average of 21 ± 6 neurons per hemisegment (mean ± SEM).

(*Gabriel et al., 2011*; *Kyriakatos et al., 2011*), while whole-cell patch-clamp recordings are obtained from V0v interneurons expressing GFP. The V0v interneurons (n = 119 from 87 preparations) were heterogeneous and displayed a range of activity patterns during fictive swimming (*Figure 2*) out of which two broad types emerged. The first includes V0v interneurons displaying rhythmic synaptic inputs coordinated with the locomotor activity recorded in the motor nerve

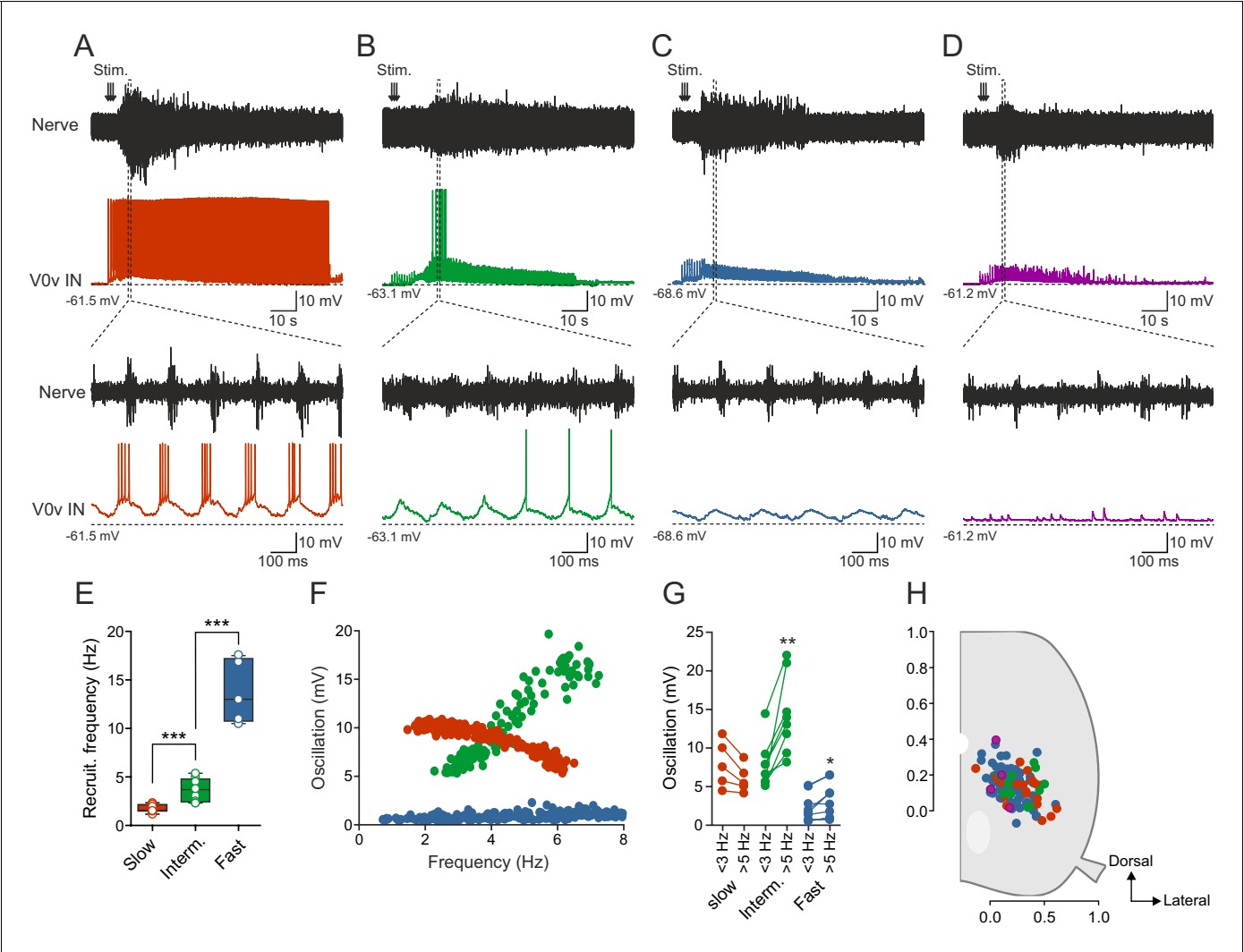

**Figure 2.** Activity of V0v interneurons during locomotion. Slow (red), intermediate (green), fast (light blue) and non-patterned (purple) V0v interneurons show different activity patterns during locomotion. (A) Recording from a V0v interneuron that was active already at a slow swimming frequency and remained recruited at high frequencies. (B) Recording from a V0v interneuron that was not active at slow swimming frequencies and became recruited only at intermediate and high frequencies. (C) Recording from a V0v interneuron that only displayed subthreshold membrane potential oscillations that did not reach the firing threshold at slow and intermediate swimming frequencies. (D) Recording from a V0v interneuron whose activity was non-patterned by swimming activity. (E) Recruitment frequency was significantly different between the three subtypes of V0v interneurons with patterned membrane potential oscillations during swimming activity (one-way ANOVA, $F_{(2, 28)}$ = 123.3, p<0.0001; followed by Tukey's procedure as *post hoc*; graph displays means and SEM; n = 17 from 16 preparations, 9 from 8 preparations and 5 from 5 preparations for slow, intermediate and fast, respectively). (F) The amplitude of the membrane potential oscillations of the three (slow, intermediate and fast) rhythmically active V0v interneurons varied with swimming frequency. (G) The membrane oscillation amplitude increased significantly between slow (<3 Hz) and intermediate (>5 Hz) swimming frequencies for the V0v interneurons recruited at intermediate and fast swimming frequencies (Student's paired t-test; p=0.035 for slow, p=0.0029 for intermediate, p=0.018 for fast; n = 5 from 5 preparations, 8 from 8 preparations, and 9 from 7 preparations for slow, intermediate and fast, respectively). (H) There is no topographic organization of the somata of the different V0v interneurons in the spinal motor column. There was no significant difference (one-way ANOVA, $F_{(3, 115)}$ = 2.129, p>0.05) between the soma size of the different sub-classes (slow: 40.4 ± 1.5; intermediate: 44.5 ± 3.0; fast: 41.8 ± 1.2; and unpatterned: 53.3 ± 9.9).

(*Figure 2A–C*), and the second comprises interneurons showing tonic and non-patterned synaptic activity (*Figure 2D*).

The V0v interneurons that displayed rhythmic activity could be further subdivided based on their activity pattern, recruitment frequency and the amplitude of their synaptic inputs. Based on these criteria, three sub-classes emerged that conform to the same organization previously found for the motor neurons and the V2a interneurons (*Gabriel et al., 2011*; *Ausborn et al., 2012*; *Ampatzis et al., 2013*). One sub-class of V0v interneurons bore characteristics previously found in motor neurons and V2a interneurons belonging to the slow microcircuit module. These neurons are recruited at the onset of the locomotor activity and fire action potentials throughout a swim episode (*Figure 2A*). These slow V0v interneurons always received suprathreshold excitation that allowed them to be recruited already at slow swimming speed (range: 1.18–2.35 Hz; 1.74 ± 0.10 Hz; *Figure 2E*), which corresponds to the lowest frequency obtained in these experiments. The second sub-class of V0v interneurons was recruited only when the swimming activity fell within an intermediate frequency span, a feature of neurons belonging to the intermediate microcircuit module (*Figure 2B*). These intermediate V0v interneurons received subthreshold synaptic excitation at slow speeds (<3 Hz) and were recruited at a minimum locomotor frequency of 3.64 ± 0.40 Hz (range: 2.33–5.39; Hz; *Figure 2E*). Finally, the third sub-class displayed only subthreshold rhythmic activity and did not fire action potentials at the slow and intermediate swimming frequencies elicited in our preparation, a characteristic feature of neurons belonging to the fast microcircuit module (*Figure 2C*). The minimum recruitment frequency of fast V0v interneurons was estimated to 13.78 ± 1.46 Hz (range: 10.50–17.60 Hz; *Figure 2E*) by linearly extrapolating the change in their synaptic inputs to their firing threshold. These neurons are likely to display a larger increase in the amplitude of the synaptic input they receive when the swim frequency approaches their recruitment threshold (see [*Ampatzis et al., 2013*]).

The three sub-classes of V0v interneurons could also be clearly distinguished based on the change in the amplitude of their rhythmic synaptic inputs as a function of the locomotor frequency (*Figure 2F*). Synaptic inputs were measured from the peak to the through of the oscillations. At locomotor frequencies below 3 Hz, the slow V0v interneurons received large synaptic inputs (red; 8.0 ± 1.35 mV) followed by the intermediate ones (green; 7.9 ± 1.2 mV) while fast interneurons received the weakest inputs (blue; 2.7 ± 0.3 mV). The amplitude of the synaptic inputs decreased somewhat for the slow, but increased in the intermediate and fast V0v interneurons when the locomotion reached intermediate frequencies (5–10 Hz) reaching an amplitude of 6.13 ± 0.8 mV in slow; 14.5 ± 2.0 mV in intermediate and 3.3 ± 0.8 mV in fast V0v interneurons (*Figure 2G*). The different functional sub-classes of V0v interneurons did not show any difference in their soma size, nor did they occupy a specific location in the spinal cord, and were hence not organized in a topographic manner (*Figure 2H*). The lack of a topographic organization is contrary to what we have previously found for the motor neurons (*McLean et al., 2007*; *Gabriel et al., 2011*), but in accordance with the V2a interneurons (*Ausborn et al., 2012*).

## Recruitment order is determined by synaptic input and intrinsic properties

Since the activity of a neuron during locomotion is very likely to be determined by the balance between its intrinsic properties and the synaptic input it receives, we decided to investigate the nature of the synaptic input to the V0v interneurons in more detail. We recorded the excitatory and inhibitory currents received by the V0v interneurons in voltage clamp at −65 mV (reversal potential of inhibition) and 0 mV (reversal potential of excitation), respectively. All V0v interneurons, except those receiving non-patterned inputs, displayed rhythmic and alternating excitatory and inhibitory currents during swimming activity (*Figure 3A–C;* n = 8 from 8 preparations for slow, n = 8 from 7 preparations for intermediate, and n = 20 from 18 preparations for fast). Current amplitude varied with swimming frequency, such that the neurons received larger currents at higher swimming frequencies (*Figure 3A–C*). This observation corresponded well with the variation of oscillation amplitude measured in current-clamp (see *Figure 2*), suggesting that neurons displaying high amplitude oscillations in current-clamp receive large excitatory and inhibitory currents. The excitatory current underlies the on-cycle excitation controlling the recruitment of the V0v interneurons, while the inhibitory current is mostly controlling the mid-cycle inhibition. Therefore, we focused our analysis on the excitatory current and how it controls the activity of the V0v interneurons at different swimming

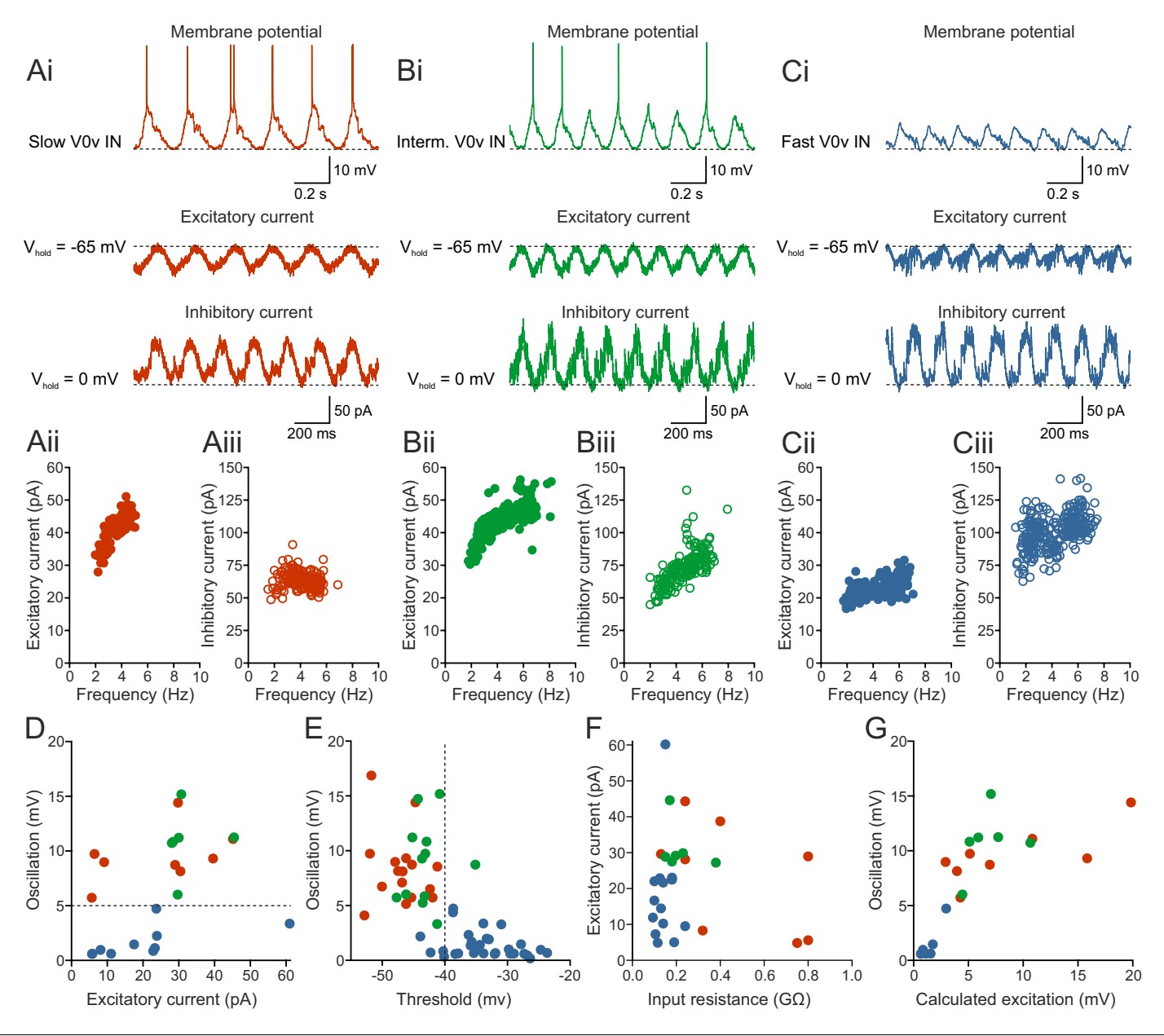

**Figure 3.** Recruitment order is determined by synaptic input and intrinsic properties. (**Ai–Ci**) The patterned activity of the V0v interneurons was mediated by rhythmic and alternating excitatory and inhibitory currents. (**Aii–Ciii**) Both the excitatory and inhibitory current amplitude was dependent on the swim frequency, but did not vary significantly between the three V0v interneuron subtypes for excitatory currents (one-way ANOVA $F_{(2,26)}$ = 2.13, p>0.05; Tukey's procedure as *post hoc*; n = 8 from 8 preparations, 6 from 5 and 15 from 15 preparations for slow, intermediate and fast respectively) and only between intermediate and fast for the inhibitory currents (one-way ANOVA $F_{(2,24)}$ = 5.21, p<0.05; Tukey's procedure as *post hoc*; n = 8 from 8 preparations, 6 from 5 preparations, and 13 from 13 preparations for slow, intermediate and fast, respectively). (**D**) There was no correlation between the amplitude of the membrane potential oscillations and the excitatory currents. Slow/intermediate neurons (red and green) differed significantly from fast neurons (blue) in oscillation amplitude (Student's t-test; p<0.0001; n = 14 and 10 for slow/intermediate and fast neurons, respectively). Slow/intermediate neurons always had oscillation amplitudes above 5 mV and fast always below 5 mV (as indicated by the dashed line). (**E**) There was a correlation between the amplitude of the membrane potential oscillations and the firing thresholds of the different V0v interneurons. Firing thresholds of slow/intermediate (red and green) neurons were significantly different from those of fast (blue) neurons (Student's t-test; p<0.0001; n = 26 and 28 for slow/intermediate and fast neurons, respectively). Most slow/intermediate neurons had a firing threshold below −40 mV and most fast neurons a firing threshold above −40 mV (indicated by the dashed line). (**F**) There was no correlation between the amplitude of the excitatory current and the input resistance of the different V0v interneurons. (**G**) The amplitude of the membrane potential oscillations was strongly correlated with the calculated excitatory drive received by the different V0v interneurons (linear regression, $R^2$ = 0.52).

frequencies. However, there was no correlation between the amplitude of voltage oscillations and excitatory synaptic current received by the three sub-classes of V0v interneurons (*Figure 3D*). Interestingly, slow and intermediate V0v interneurons could be segregated from the fast interneurons based on the amplitude of rhythmic inputs (voltage oscillations) they received during locomotion, the former always displaying oscillation amplitudes above 5 mV at slow/intermediate swimming frequencies (i.e., 4–5 Hz; *Figure 3D*). In addition, there was also a separation between the firing thresholds of slow/intermediate and the fast V0v interneurons at around −40 mV (*Figure 3E*). These results indicate that the recruitment of the different V0v interneurons is not solely defined by the excitatory drive they receive, but also by their firing properties.

Furthermore, the order of recruitment of the different V0v interneurons did not correlate with the input resistance since there was a large overlap of the input resistance of the three V0v interneuron subclasses. In addition, no correlation was found between the amplitude of the excitatory current and the input resistance of V0v interneurons (*Figure 3F*). These results indicate that neither the excitatory current nor the input resistance alone can account for the resulting rhythmic excitation setting the recruitment order of the V0v interneuron subclasses. Another possibility is that the synaptic inputs received by the V0v interneurons during swimming results from a combined contribution of their excitatory currents and their input resistance. In this scenario, each sub-class would always receive optimal excitation by compensating for a weak current with a high input resistance or a strong current with a low input resistance. To determine if this is the case, we calculated the excitation received by each V0v interneuron by multiplying the measured excitatory currents with the input resistance of the neurons. There was a strong correlation between the calculated excitation and the measured rhythmic inputs recorded during swimming in the different V0v interneurons (*Figure 3G*; $R^2$ = 0.52; p<0.001). This indicates that the amount of excitation received by each V0v interneuron is the result of a scaling of the excitatory current and input resistance. These results show that the recruitment of the V0v interneurons is determined by a combination of their intrinsic properties and the received excitatory drive.

## V0v interneuron sub-classes display distinct intrinsic properties

To determine if the different sub-classes of V0v interneurons display specific firing patterns, current pulses of different amplitudes were applied and the resulting active responses examined. The slow V0v interneurons displayed a tonic or bursting firing pattern with little or no adaptation (*Figure 4A*). The intermediate V0v interneurons also displayed a tonic or bursting firing pattern (*Figure 4B*). However, the fast and non-patterned V0v interneurons displayed adaptation and in some cases only fired maximally a single action potential in response to the injected depolarizing current steps (*Figure 4C, D*). In addition, the firing thresholds of the different V0v interneurons were graded in a manner that conforms to their order of recruitment during locomotion. The slow V0v interneurons display the lowest thresholds, followed by the intermediate and finally the fast and the non-patterned V0v interneurons. Thresholds were −46.2 ± 0.7 mV; −43.0 ± 1.0 mV; −35.2 ± 0.6 mV; and −29.5 ± 3.4 mV for slow, intermediate, fast and non-patterned neurons, respectively (*Figure 4E*).

In order to confirm the categorization of the rhythmically active V0v interneurons into slow, intermediate and fast based on their intrinsic properties, a principal components analysis was performed. This analysis allowed several parameters (input resistance, action potential threshold, membrane oscillation amplitude in current clamp, and minimum recruitment frequency) to be taken into account at once, creating a multidimensional view of the data. There was a clear separation of the fast V0v interneurons from the slow and intermediate. The slow and the intermediate V0v interneurons could also be partly segregated with some overlapping parameters (*Figure 4G*), concurring with their differential activation during locomotion. Thus, the rhythmically active V0v interneurons can be separated into the three different groups previously described for motor neurons and V2a interneurons.

## Morphological diversity of V0v interneurons

The morphological identity of V0v interneurons has been previously examined in zebrafish larva. By the use of birth date analysis, it was found that the V0v interneurons could be divided into three morphologically distinct groups with ascending, descending and bifurcating axonal projections. The latter two groups arise later from common progenitors whereas the former is produced the earliest and from distinct progenitors (*Satou et al., 2012*). Additionally, it has been suggested that earlier

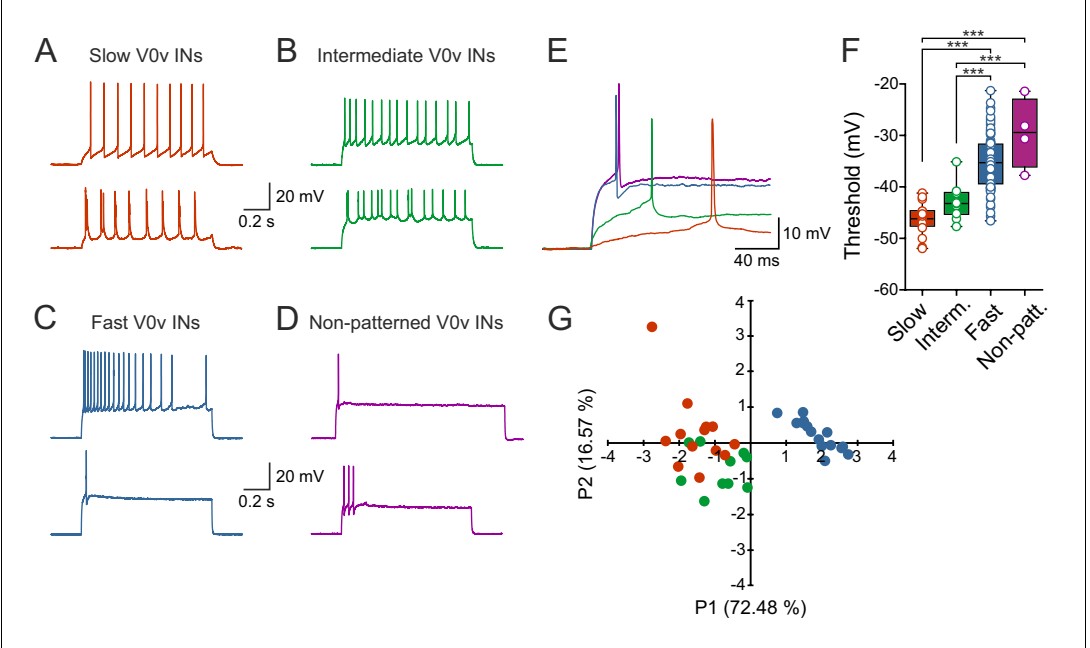

**Figure 4.** V0v interneuron sub-classes display distinct intrinsic properties. (A–D) Examples of two different neurons from each group firing in response to an injected subthreshold current step (A) The V0v interneurons active already at slow swimming frequencies were either tonically firing or fired in bursts in response to current injections. (B) The V0v recruited at intermediate frequencies also displayed tonic firing or bursting in response to current injections. (C) The fast V0v interneurons always displayed strong adaptation. (D) The V0v interneurons with non-patterned changes in the membrane potential fired only a few action potentials and displayed strong adaptive properties. (E) The different V0v interneurons had different thresholds for firing action potentials in response to injected current. (F) The firing thresholds were significantly different from each other and correlated strongly with the recruitment order as a function of swimming frequency (one-way ANOVA $F_{(3,104)}$ = 33.8, p<0.0001; Tukey's procedure as *post hoc*; n = 19 from 17 preparations, 11 from 11 preparations, 74 from 58 preparations, and 4 from 4 preparations for slow, intermediate, fast and non-patterned respectively). (G) A principal component analysis revealed that the V0v interneurons with patterned membrane potential oscillations could be separated from each other with those active at slow and intermediate frequencies showing partial overlap. The percentage of the total variance explained by PC1 and PC2 were 72.48 and 16.57, respectively. The percentage contribution of the different variables to PC1 and PC2, respectively, was: input resistance 20.4 and 44.9; action potential threshold 26.5 and 4.1; membrane oscillation amplitude in current clamp 21.5 and 48.6; minimum recruitment frequency 31.6 and 2.3 (n = 13 neurons from 13 preparations, 10 neurons from 9 preparations, and 15 neurons from 13 preparations for slow, intermediate and fast, respectively).

born spinal neurons in the larval zebrafish form the fast system, whereas those later born are involved in slow swimming (*Liu and Westerfield, 1988*; *Kimura et al., 2006*; *McLean and Fetcho, 2009*; *Fetcho and McLean, 2010*). Taken together, these studies suggest that fast V0v interneurons would predominantly be of the ascending morphology, whereas the intermediate and slow V0vs would be predominantly bifurcating and descending. The morphologies of V0v interneurons in the adult zebrafish could be analyzed post-experimentally by using neurobiotin in the patch pipette (*Figure 5*). Contrary to what the studies in larval zebrafish suggest, no correlation was found between the morphology of a neuron and its activity pattern during locomotion (*Figure 5A–D*). All neurobiotin-filled neurons were confirmed to be commissural neurons and a large proportion of them correspond to those that would be recruited at faster swimming speeds (*Figure 5E*). In addition, the majority of V0v interneurons across all four groups had descending axons. These axons could be between 2 and 22 segments long and both short and long axons were found in all four groups. Apart from descending, we also found ascending, bifurcating and local neurons. Bifurcating neurons were defined as having axonal projections stretching over one segment in both ascending and descending directions and local neurons were defined as having projections not extending into the adjacent segment of that where the soma was located. Moreover, the descending and bifurcating V0v interneurons could either have axonal projections that were smooth, or extending multiple collaterals, mainly in the dorsal and rostro-caudal directions (*Figure 5D*). Neurons from all four groups displayed extensive dendritic trees in contrast to what has been documented in larvae (*Satou et al.,*

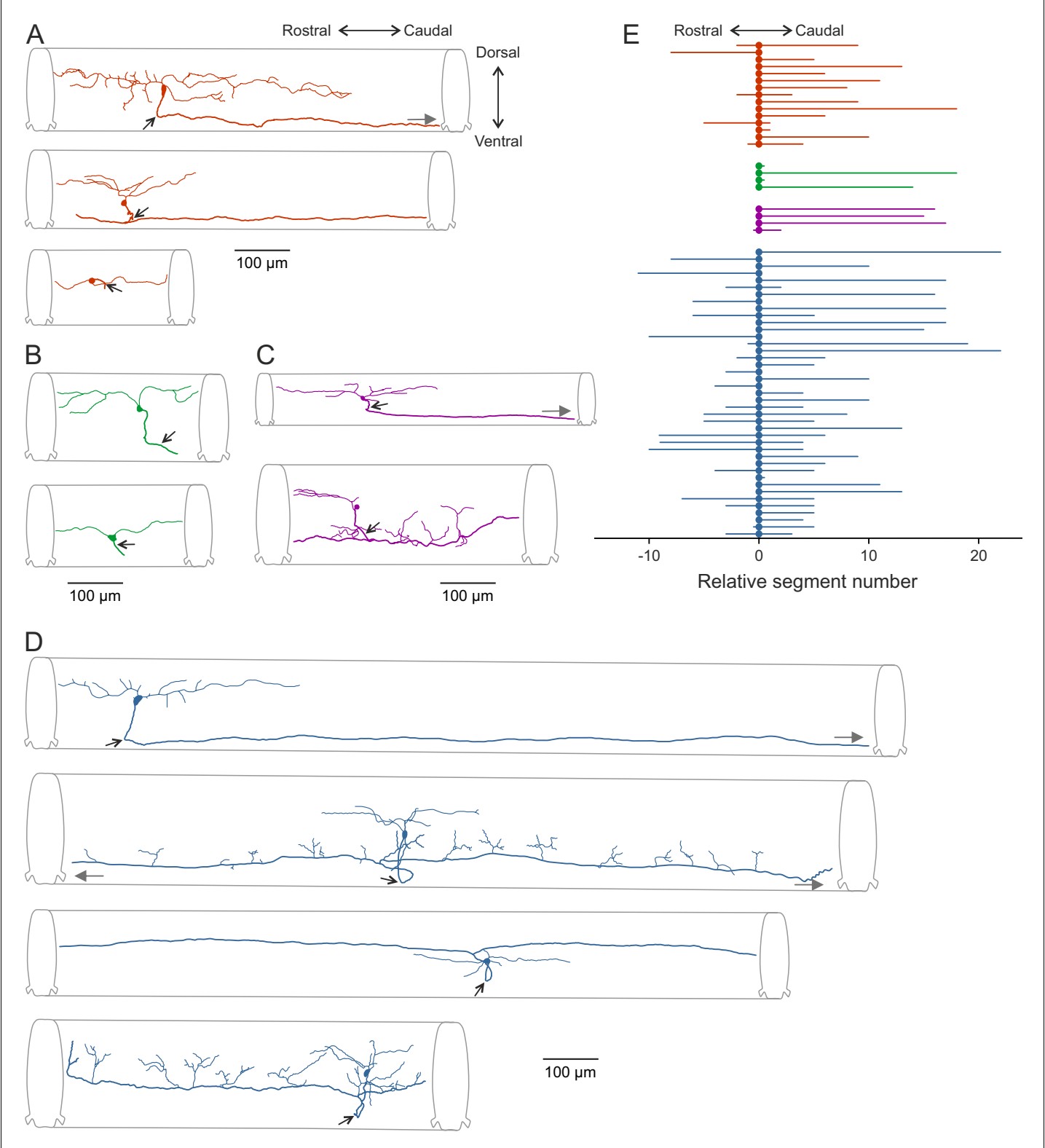

**Figure 5.** Morphological diversity of V0v interneurons. (A–D) Examples of dye-labeled V0v interneurons. (A) Slow, (B) intermediate, (C) non-patterned, (D) fast. The morphologies of the different V0v interneurons show heterogeneity in terms of their axonal and dendritic projections. Small, black arrows indicate where the axon crosses to the contralateral side; grey, filled grey arrows indicate that the axon projects further than illustrated (E) Graph showing the range of the axonal projections of the different neurobiotin-labeled V0v interneurons.

*2012*), suggesting rather large developmental changes in the morphology of neurons between larvae and adults. It is worth noting that none of the V0v interneurons had the distinct morphological characteristic of the MCoDs, perhaps suggesting that these V0v interneurons are a larval feature or undergo dramatic morphological remodeling as the network matures into that of the adult.

## Discussion

### Functional heterogeneity of commissural excitatory interneurons in adult zebrafish

This study reveals that the excitatory commissural V0v interneurons represent a functionally heterogeneous class. Although they express a defined transcription factor and transmitter phenotype, their activity pattern during locomotion is not uniform. This interneuron class could be segregated into two distinct types based on whether they display rhythmic activity or not during locomotion. The rhythmically active V0v interneurons could be further subdivided into three main sub-classes engaged sequentially at slow then intermediate and finally fast locomotor speed. The order of recruitment of these three sub-classes of V0v interneurons is defined by the integration of their synaptic drive with their intrinsic properties. The extent of the rhythmic excitation results from a scaling of the synaptic current with the input resistance of the different V0v interneurons. This study thus provides a systematic characterization of the V0v interneurons showing that they represent a functionally heterogeneous population with different sub-classes engaged in a speed-dependent manner during locomotion.

### Development and perturbations of excitatory commissural interneurons in vertebrates

The molecular mechanisms underlying the development and differentiation of V0 commissural interneurons seem to be conserved between zebrafish and mouse (reviewed in [*Goulding, 2009*]). These interneurons can be segregated into a dorsal inhibitory (V0d), and a ventral excitatory population (V0v). The progenitors giving rise to these interneurons express the transcription factor Dbx1 the elimination of which altered the left-right coordination (*Pierani et al., 2001*; *Lanuza et al., 2004*; *Talpalar et al., 2013*). Selective ablation of V0d and V0v interneurons in the immature mouse suggested a differential contribution depending on the speed of locomotion (*Talpalar et al., 2013*; *Kiehn, 2016*). Based on these results, it has been suggested that V0d and V0v interneurons act as two homogenous classes controlling the left-right alternation in a speed-dependent manner with a switch from V0d to V0v as the locomotor activity changes from slow to fast frequencies. However, no recordings have thus far been made from identified V0d or V0v interneurons to assess directly whether they do indeed represent a homogenous class and how their activity patterns change as a function of the locomotor frequency.

A detailed analysis of the development of V0v interneurons in zebrafish has revealed that these commissural excitatory interneurons consist of three sub-classes that differentiate in a specific temporal order correlating with their axonal trajectories (*Satou et al., 2012*). Despite the many similarities and conserved developmental mechanisms of V0 interneurons between zebrafish and mice, the available results also point out some apparent discrepancies. In addition, the increase in locomotor speed in mice can be associated with change in gait while in fish there is no change in gait and the left-right alternation is maintained at all speeds. In mice both V0d and V0v play a role in ensuring the left-right alternation. The changes in the locomotor pattern following ablation of V0v interneurons in immature mice suggest that they act via di-synaptic inhibition by recruiting contralateral inhibitory interneurons at relatively faster speeds. In contrast, recordings from V0v MCoDs in larval zebrafish show that these neurons are only active during slow swimming (*McLean et al., 2007*; *McLean et al., 2008*; *Fidelin et al., 2015*). Our results now show, in an adult vertebrate, that the V0v interneurons are more functionally diverse. Some are engaged at slow swimming while others are active at intermediate and fast swimming. The fact that a large proportion of V0v interneurons recorded in our study belong to the fast category, suggests that the V0v interneurons are primarily recruited at faster locomotor speeds.

Our work indicates some possible developmental changes in the zebrafish V0v interneurons. In addition to the prevalence of fast V0v interneurons, we found no evidence for interneurons with

morphology resembling the MCoDs described in larval zebrafish, although V0v interneurons active during slow frequency swimming were recorded from in our study. The absence of MCoDs in the adult zebrafish could result from the unsuccessful recombination in some V0v interneurons. Alternatively, there could be a downregulation of the GFP expression in these neurons. However, in larvae, the MCoDs are responsible for driving slow frequency locomotion (*McLean et al., 2008*), whereas in the adult zebrafish this task appears to be carried out largely by slow V2a interneurons (*Ausborn et al., 2012*). The results clearly show a heterogeneity in both the V0v interneuron activity pattern and morphology, arguing against the downregulation of GFP or unsuccessful recombination.

### The functional role of V0v interneurons during locomotion

Our results demonstrate that the V0v interneurons that are active during locomotion can be subdivided into slow, intermediate and fast subclasses recruited gradually as swimming speed increases. This subdivision matches our previous findings of a similar segregation of motor neurons and the excitatory V2a interneurons in adult zebrafish (*Gabriel et al., 2011*; *Ausborn et al., 2012*; *Ampatzis et al., 2013*). Here, the locomotor circuit is composed of three sub-circuits of connected V2a interneurons and motor neurons engaged sequentially from slow to intermediate and then fast swimming. We now show that the same organization also applies to the V0v interneurons, which represent an important population of commissural interneurons in the spinal cord. However, in our study, we did not find any clear correlation between the morphology of the V0v interneurons and their order of recruitment. It seems that there is no direct relationship between the order of differentiation of these interneurons and their functional engagement during swimming in the adult zebrafish. Although slow, intermediate and fast V0v interneurons could be clearly distinguished and identified, there was a larger proportion of fast compared to slow or intermediate V0v interneurons in our random sampling. If this is a true representation of the subdivision of the V0v interneurons, it would tend to agree with the assumption put forward in the mouse that they are mostly, albeit in zebrafish not exclusively, active at fast locomotion.

Moreover, there was an additional type of V0v interneuron not receiving rhythmic inputs during locomotion and whose function in the spinal circuitry is still unclear. There is a possibility that these non-patterned neurons are involved in fin movements rather than axial muscle movements. In lamprey, a proportion of excitatory commissural neurons have been shown to contact dorsal fin motor neurons (*Mentel et al., 2008*). Alternatively, they could be involved in postural control, or could be involved in even more vigorous movement such as struggling.

In conclusion, our study provides a systematic description of the functional diversity of V0v interneurons in an adult vertebrate. These interneurons are organized in a modular fashion and although they seem to contribute at all locomotor frequencies, those active at fast speed seem to be overrepresented in our recordings. Thus, we uncover an important organizational principle for an important class of commissural interneurons and the underlying cellular and synaptic mechanisms defining their pattern of recruitment as a function of locomotor speed.

## Materials and methods

### Animals and ex-vivo spinal cord preparation

Zebrafish (*Danio rerio*) were raised and kept in a core facility at the Karolinska Institute according to established procedures. In order to target the excitatory V0 interneurons specifically, two transgenic zebrafish lines were crossed, Tg[*dbx1b*:Cre] and Tg[*vglut2a*:loxP-DsRed-loxP-GFP] which have previously been described (*Satou et al., 2012*, *2013*). This cross gave offspring in which all dbx1b and vglut2a positive neurons expressed GFP, allowing for specific targeting for patch clamp recordings. 7–10 weeks old juvenile/adult zebrafish of both sexes were used for all experiments. All experimental protocols were approved by the local Animal Research Ethical Committee, Stockholm. The preparation was carried out as described previously (*Gabriel et al., 2011*; *Kyriakatos et al., 2011*). Zebrafish were cold-anesthetized in frozen fish saline containing in mM: 134 NaCl, 2.9 KCl, 2.1 CaCl$_2$, 1.2 MgCl$_2$, 10 Hepes and 10 glucose with pH adjusted to 7.8 using NaOH and osmolarity adjusted to 290 mOsm. The skull was opened and the brain was cut at the level of the midbrain. The epaxial musculature was removed up to the caudal end of the dorsal fin and the skin was removed from the remaining tail musculature. The vertebral arches at the first two segments of the spinal

cord were removed to allow access for extracellular stimulation of descending axons. Vertebral arches were also removed over 4–5 segments rostral to the dorsal fin position, to allow access for patch-clamp recording electrodes. The preparation consisting of the spinal cord in the vertebral column, the hindbrain and the caudal musculature were cut out and transferred to the recording chamber, placed on the side, and fixed with Vaseline. During experiments, the preparation was continuously perfused with oxygenated fish saline at room temperature, containing 10 µM D-tubocurarine (Sigma-Aldrich, Sweden) to block the neuromuscular junctions allowing for stability during patch-clamp recordings.

## Anti-GFP staining

Fish were anesthetized with 0.1% MS-222 (Sigma-Aldrich) and fixed in 4% PFA (vol/vol; Sigma-Aldrich) overnight at 4°C. The spinal cords were dissected out and washed 2 times for 10 min each in phosphate buffered saline (PBS; 0.1 M, pH 7.4, Ambion, Naugatuck, CT). Spinal cords were then incubated for 40 min at room temperature in blocking solution containing 5% bovine serum albumin (BSA, Sigma-Aldrich), 0.15% normal rabbit serum (NRS, Sigma-Aldrich), and 1% PBS-triton (x-100). Spinal cords were then incubated in rabbit anti-GFP antibody (Life Technologies, Sweden), 1:200 in 5% BSA and PBS-triton (x-100) for 48 hr at 4°C. The cords were washed 4 times for 10 min each in 1% PBS-triton (x-100) and subsequently incubated in Alexa Fluor-488 anti-rabbit IgG antibody (Life Technologies) 1:400 in PBS-triton (x-100) for 4 hr at room temperature. Spinal cords were then washed 5 times for 5 min each and 15 min in PBS and subsequently mounted with either lateral or dorsal side up in antifade fluorescent mounting medium (Vectashield, Vector Labs, Sweden). Confocal images were obtained with a Zeiss LSM 510 laser-scanning confocal microscope. The pictures were put together in CorelDraw and color was added by tracing the neurons in Photoshop for clarification purposes. Number of neurons per hemisegment were counted from confocal pictures taken at different rostro-caudal positions along the spinal cord in 6 different preparations stained for GFP. Dorso-ventral soma position was calculated from the same confocal pictures.

## Electrophysiology

Extracellular recording and stimulation electrodes were pulled from borosilicate glass (1-mm o.d., 0.87-mm i.d.; Harvard Apparatus, Holliston, MA), broken down to the desired tip diameter (15–25 µm), and fire-polished. Extracellular recordings were made from the motor nerves running through the intermyotomal clefts at the tail, where the musculature was left intact. A stimulation electrode was placed dorsally on the rostral spinal cord to elicit locomotor episodes by electrical stimulation (1 s at 30 Hz or 10 s at 1 Hz; pulse width, 1 ms; current amplitude, 0.3–1 mA). Extracellular signals were amplified (gain, 10,000) with a differential AC amplifier (A-M Systems) and filtered with low and high cutoff frequencies of 300 Hz and 1 kHz, respectively. Intracellular whole-cell recordings were performed from GFP-labelled V0v interneurons in the mid-body region rostral to the dorsal fin and peripheral nerve recording. For intracellular recordings, electrodes were pulled from borosilicate glass (1.5-mm o.d., 0.87-mm i.d.; Hilgenberg, Germany) on a horizontal puller (Sutter Instruments, Novato, CA) and filled with intracellular solution containing in mM: 120 K-gluconate, 5 KCl, 10 Hepes, 4 $Mg_2ATP$, 0.3 $Na_4GTP$, 10 Na-phosphocreatine with pH adjusted to 7.4 with KOH and osmolarity to 275. GFP-labeled V0v interneurons were visualized with a fluorescent microscope (Axioskop FS Plus; Zeiss, Germany) equipped with IR-differential interference contrast (DIC) optics and a CCD camera with frame grabber (Hamamatsu, Japan). Intracellular patch-clamp electrodes were advanced into the exposed portion of the spinal cord through the meninges using a motorized micromanipulator (Luigs & Neumann, Germany) while applying constant positive pressure. Intracellular signals were amplified with a MultiClamp 700B intracellular amplifier (Molecular Devices, Sunnyvale, CA) and low-pass filtered at 10 kHz. In current-clamp recordings, no bias current was injected. Only V0v interneurons with stable membrane potentials, which fired action potentials in response to suprathreshold depolarizations, and that showed minimal changes in series resistance (<5%) were included in this study.

## Analysis of neuron size and position

Images of the outline of the soma as well as the dorsal edge of the spinal cord and the Mauthner axon were captured before patching. The soma size and position were analyzed post-experiment

using the measurement functions in ImageJ (http://rsb.info.nih.gov/ij). For the ventrodorsal soma position, the dorsal edge of the spinal cord was defined as 1 and the dorsal edge of the Mauthner axon as 0 (*Figure 2H*). The mediolateral position of the soma was measured during the experiment by focusing between the soma position, the lateral edge of the Mauthner axon, and the lateral surface of the spinal cord. For the mediolateral soma position, the lateral edge of the Mauthner axon was defined as 0 and the lateral surface of the spinal cord as 1. All positions and soma sizes were averaged from at least three individual measurements.

### Neurobiotin histochemistry

Neurons were passively filled with neurobiotin tracer (0.25% vector labs) during patch clamp experiments to reveal their morphological characteristics post hoc. Post-recording, preparations were transferred to 4% PFA solution and left for 48–72 hr at 4°C. The spinal cords were dissected out of the vertebral column and washed 4 times for 10 min each in PBS, then incubated in 0.5–1% PBS triton for 3–4 hr and subsequently incubated in streptavidin conjugated to Alexa Fluor 555 (1:500; Invitrogen, Sweden) in PBS containing 0.5% Triton X-100 (Sigma-Aldrich) and 2% normal goat serum (Sigma-Aldrich) 24–72 hr at 4°C. The tissue was then washed in PBS 3 times for 10 min each and 2 times for 20 min each and mounted in antifade fluorescent mounting medium (Vectashield, Vector Labs). Confocal images were obtained with a Zeiss LSM 510 laser-scanning confocal microscope. The pictures were put together and colored in Photoshop and CorelDraw.

### Data acquisition and analysis

Data were digitized at 10 kHz (extracellular recordings) or 20 kHz (patch clamp recordings) with a Digidata 1322A A/D converter (Molecular Devices) and acquired using pClamp software (version 10; Molecular Devices). Data analysis was performed in Matlab 2011a, Clampfit software and Axograph version 1.5.4. The action potential voltage threshold of each V0 interneurons was determined from the measured membrane potential at which the *dV/dt* exceeded 10 mV/ms. The input resistance of neurons was calculated as the slope of a regression line to the linear region of the I–V curve (membrane potentials of −80 to −60 mV), which was obtained by injection of hyperpolarizing current pulses (duration, 1–2 s). The minimum recruitment frequency of V0 interneurons was defined as the lowest locomotor frequency of a swimming episode at which the neurons were firing action potentials. The minimum locomotor frequency obtained during swimming episodes varied between preparations, therefore, the minimum recruitment frequency of the slow V0v interneurons will be the minimum frequency obtained from those preparations. To analyze the postsynaptic currents (PSCs), V0 interneurons were voltage clamped close to the reversal potential of excitation (0 mV) or inhibition (−65 mV). The peak-to-trough amplitude of the summed excitatory and inhibitory currents underlying the locomotor-driven membrane potential oscillations were analyzed and not the individual PSCs. Graphs and statistical analysis were made with GraphPad Prism 4. Correlation analyzes were carried out using linear regressions. For normally distributed data, Student's t-test (two groups) or one-way ANOVAs (> two groups) were used with Tukey's procedure as *post-hoc*. Confidence intervals were 95%. Gaussian fits were carried out on all data. None of the analyzed data was abnormally distributed. Differences were considered to be significant if $p < 0.05$. All data are given as mean ± SEM. The PCA was carried out using xlStat. Factors were included in the PCA if they had a value significantly different from zero using a Pearson correlation matrix. Factors included were input resistance, action potential threshold, membrane potential oscillation amplitude and minimum recruitment frequency. Figures were put together in CorelDraw.

## Acknowledgements

We thank Dr. Keith Sillar and members of our lab for comments on an early version of the manuscript. We are grateful to Shin-ichi Higashijima for providing the zebrafish lines used in this study. This study was supported by a grant from the Swedish Research Council, Karolinska Institute and the Swedish Brain Foundation.

## Additional information

### Funding

| Funder | Author |
|---|---|
| Karolinska Institutet | Abdeljabbar El Manira |
| Swedish Brain Foundation | Abdeljabbar El Manira |
| Swedish Research Council | Abdeljabbar El Manira |

The funders had no role in study design, data collection and interpretation, or the decision to submit the work for publication.

### Author contributions

ERB, Conception and design, Acquisition of data, Analysis and interpretation of data, Drafting or revising the article; AEM, Conception and design, Analysis and interpretation of data, Drafting or revising the article

### Author ORCIDs

Abdeljabbar El Manira, http://orcid.org/0000-0001-5920-9384

### Ethics

Animal experimentation: All experimental protocols were approved by the local Animal Research Ethical Committee, Stockholm (Stockholms Norra Djurförsöksetiska Nämnd). Protocol number N122/13.

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
