## [Decision Letter]

Thank you for submitting your article "Functional diversity of excitatory commissural interneurons in adult zebrafish" for consideration by *eLife*. Your article has been reviewed by three peer reviewers, one of whom is a member of our Board of Reviewing Editors and the evaluation has been overseen by Eve Marder as the Senior Editor. The following individuals involved in review of your submission have agreed to reveal their identity: Ronald L Calabrese (Reviewer #1 and Reviewing Editor); Rob Brownstone (Reviewer #2); Wen-chang Li (Reviewer #3).

The reviewers have discussed the reviews with one another and the Reviewing Editor has drafted this decision to help you prepare a revised submission.

Summary:

In this interesting study, the authors perform an electrophysiological and anatomical analysis of V0v glutamatergic commissural neuron in the adult zebrafish. They find that these neurons are anatomically and functionally diverse; the activity pattern during locomotion consist of two distinct types dependent on whether they display rhythmicity or not during locomotion. The rhythmic V0v interneurons could be further subdivided into three sub-classes engaged sequentially, first at slow then intermediate and finally fast locomotor speeds. Their order of recruitment is defined by the combination of their synaptic current and their input resistance. There is no apparent anatomical or positional correlation with activity type. The results contrast markedly with finding in the larval fish and illuminate the similar studies in the mouse. Thus these findings have significant implications for how side-to-side coordination is effected during locomotion in adult vertebrates.

Essential revisions:

1) The interpretation of the manuscript relies on accurate sub-classification of the neurons. This seems somewhat arbitrary in the first instance, with the PCA being brought in later. The data may be more convincing if the initial classification were based on a well described, thorough PCA. At minimum, it should be explicit that the cut-offs between groups are arbitrary.

The sub-categorization of the rhythmically active V0v interneurons seems based on a frequency scheme previously developed by the lab for other neurons, and then supported post-hoc in part by the PCA. P5, last 9 lines reads that the authors have arbitrarily set some cut-off recruitment frequencies for the V0v subclasses. It is acceptable to set some criteria for grouping and description purposes, but this should not be confused with results. In this case, the frequency ranges and averages should be given in this part of text instead of being described as results in Figure 2 later with stats.

Could the PCA be used more up front to rationalize this categorization rather than as a post hoc justification? If anatomical factors (e.g., soma size, axon projections and lengths, dendritic branching) are used in the PCA can it better discriminate the three main subcategories of rhythmic V0v neurons?

No matter whether an expanded PCA is used or the original, the PCA must be better described. Identify the factors clearly in Methods: input resistance, action potential threshold, membrane oscillation amplitude in current clamp, and minimum recruitment frequency. Even the text does not make explicitly clear that these were the only factors considered. For each of the two principal components illustrated in Figure 4 you should list the factors with their relative coefficients, so that we can interpret the data properly. Are all factors equally weighted or is one in particular outstanding for each component? In Methods, at least, all this information should be provided. In Figure 4 you give% for the first 2 components but you never state that this is the% variance explained. The Figure 4 caption describing this analysis is inadequate.

2) Inhibition is barely described in the text associated with Figure 3. Please tells us what we are to learn from the analyses of inhibition and discuss its significance.

3) More information on the longitudinal location of V0v neurons should be provided. The authors state V0v neurons are evenly distributed along the rostral-caudal axis. However, the density of GFP neurons in Figure 1 look quite different from those in Figure 1. Can a plot of r-c location against dorsal-ventral measurements of somata can be presented to replace Figure 1. This is also relevant to Figure 5, where the r-c information of soma position is missing. One may intuitively think that more caudal neurons have a higher chance of possessing ascending branches while the rostral ones tend to have descending branches. To register the r-c locations in the results could clearly show if there is such a relation. We understand that there might be some favorite locations for whole-cell recordings for the experimenter. The inclusion of such information, however, can help to draw more accurate conclusions.

Because of the anatomical and functional diversity found for the V0v interneurons, does it make sense to argue in Discussion that axonal morphology reflects birth order; isn't it just as likely that there are anatomical transformations associated with moving from larva to adult that make the correlation impossible to make?

4) Introduction is confusing. It starts talking about animal locomotion but without further indication jumps into vertebrates with limbs and then jumbles fish and mice; limbed and non-limed vertebrates. The authors should make the logical flow clear. The reference to changes in gait from alternation to hopping with speed are confusing in the context of the fish. Fish change speeds but not gaits; one always expects side-to-side alternation regardless of speed. This should be made clear.

The Discussion too needs to make it clear that charges of speed in the fish are not accompanied by changes in side-to-side coordination.

5) Results related to Figure 3. The input resistance of most neurons described here are lower than 400 MOhm. One would expect a correlation between synaptic current size and oscillation amplitudes. There is a clear outlier on Figure 3. Have you tried correlation with that rightmost data point omitted?

---

## [Author Response]

*1) The interpretation of the manuscript relies on accurate sub-classification of the neurons. This seems somewhat arbitrary in the first instance, with the PCA being brought in later. The data may be more convincing if the initial classification were based on a well described, thorough PCA. At minimum, it should be explicit that the cut-offs between groups are arbitrary.*

*The sub-categorization of the rhythmically active V0v interneurons seems based on a frequency scheme previously developed by the lab for other neurons, and then supported post-hoc in part by the PCA. P5, last 9 lines reads that the authors have arbitrairily set some cut-off recruitment frequencies for the V0v subclasses. It is acceptable to set some criteria for grouping and description purposes, but this should not be confused with results. In this case, the frequency ranges and averages should be given in this part of text instead of being described as results in Figure 2 later with stats.*

We have now clarified that the three sub-classes of V0v interneurons was based on the pattern of their activity during swimming, their recruitment frequency and the amplitude of the synaptic inputs they receive (subsection “Activity pattern of glutamatergic V0 interneurons during locomotion”). In addition, we have also added the frequency ranges and averages as suggested by the reviewers.

Could the PCA be used more up front to rationalize this categorization rather than as a post hoc justification?

Subject to your approval, we prefer not to use the PCA more up front as suggested by the reviewers since this would break the flow of the results. However, we have modified the text to explicitly state the criteria used to categorize the three sub-classes of the V0v interneurons.

*If anatomical factors (e.g., soma size, axon projections and lengths, dendritic branching) are used in the PCA can it better discriminate the three main subcategories of rhythmic V0v neurons?*

Soma size has been analyzed for the four different groups and it was not significantly different. The numbers are stated in the legend to Figure 2. Furthermore, when soma size was taken into consideration for the principal components analysis, it did not improve the discrimination of the sub-classes. Axonal projections and lengths could not be added to the PCA since all the recorded neurons could not be recovered for detailed morphological analysis. Figure 5 shows that the different sub-classes displayed heterogeneous axonal projections and length.

*No matter whether an expanded PCA is used or the original, the PCA must be better described. Identify the factors clearly in Methods: input resistance, action potential threshold, membrane oscillation amplitude in current clamp, and minimum recruitment frequency. Even the text does not make explicitly clear that these were the only factors considered. For each of the two principal components illustrated in Figure 4 you should list the factors with their relative coefficients, so that we can interpret the data properly. Are all factors equally weighted or is one in particular outstanding for each component? In Methods, at least, all this information should be provided. In Figure 4 you give% for the first 2 components but you never state that this is the% variance explained. The Figure 4 caption describing this analysis is inadequate.*

We now describe the PCA in details and included the list of the factors with their relative coefficients. We have also changes the legend of Figure 4 to indicate the percent variance explained by the first two components.

*2) Inhibition is barely described in the text associated with Figure 3. Please tells us what we are to learn from the analyses of inhibition and discuss its significance.*

We have now added information about the amplitude of the inhibitory current received by the different V0v interneurons sub-classes and discussed briefly the significance of the inhibition (subsection “Recruitment order is determined by synaptic input and intrinsic properties”). Our analysis focuses on the mechanisms directly controlling the recruitment of the different V0v interneurons and therefore more emphasis was given to the excitatory current underlying the on-cycle depolarization.

*3) More information on the longitudinal location of V0v neurons should be provided. The authors state V0v neurons are evenly distributed along the rostral-caudal axis. However, the density of GFP neurons in Figure 1 look quite different from those in Figure 1. Can a plot of r-c location against dorsal-ventral measurements of somata can be presented to replace Figure 1.*

The difference in density of GFP neurons in Figure 1 compared to Figure 1 is mainly due to the amount of depth provided in the pictures. For Figure 1, 164 and 193 stacks, respectively, were included in the projections to obtain the most information about the number of neurons. In Figure 1 only 90 stacks were included to clearly show the axonal projections illustrating the commissural processes of two example neurons. Information about the images has been added to the figure legend for clarity. A new graph (Figure 1) has been added to show the dorso-ventral and the rostro-caudal distribution of V0v interneurons within one segment from three different preparations.

*This is also relevant to Figure 5, where the r-c information of soma position is missing. One may intuitively think that more caudal neurons have a higher chance of possessing ascending branches while the rostral ones tend to have descending branches. To register the r-c locations in the results could clearly show if there is such a relation. We understand that there might be some favorite locations for whole-cell recordings for the experimenter. The inclusion of such information, however, can help to draw more accurate conclusions.*

We have added to the Materials and methods that patch-clamp recordings were made from a restricted region in the mid-body of the animal. No recordings were made from the rostral most or the caudal most parts of the spinal cord. The recorded V0v interneurons show a heterogeneous axonal projection pattern across the different sub-classes although they were all recorded within a restricted spinal cord region (see Figure 5).

*Because of the anatomical and functional diversity found for the V0v interneurons, does it make sense to argue in Discussion that axonal morphology reflects birth order; isn't it just as likely that there are anatomical transformations associated with moving from larva to adult that make the correlation impossible to make?*

We have revised the Discussion to clarify this point.

4) Introduction is confusing. It starts talking about animal locomotion but without further indication jumps into vertebrates with limbs and then jumbles fish and mice; limbed and non-limed vertebrates. The authors should make the logical flow clear. The reference to changes in gait from alternation to hopping with speed are confusing in the context of the fish. Fish change speeds but not gaits; one always expects side-to-side alternation regardless of speed. This should be made clear.

We have modified the Introduction to make the logical flow clear. We have also made clear that in mice both V0d and V0v interneurons underlie the left-right alternation. The excitatory V0v are thought to mediate indirect inhibition of neurons on the contralateral side by activating inhibitory interneurons. This is now clarified in the Introduction.

*The Discussion too needs to make it clear that charges of speed in the fish are not accompanied by changes in side-to-side coordination.*

We now make clear in the Discussion that changes of speed in fish are not associated with changes in gait.

*5) Results related to Figure 3. The input resistance of most neurons described here are lower than 400 MOhm. One would expect a correlation between synaptic current size and oscillation amplitudes. There is a clear outlier on Figure 3. Have you tried correlation with that rightmost data point omitted?*

The correlation between the locomotor-related oscillations and excitatory currents included all the data points. The neuron in Figure 3 has an input resistance lower than 400 MOhm and is not an outlier for any other parameter measured than the excitatory current. Hence, we have chosen to include it in the analysis.